# QGAN: QUANTIZE GENERATIVE ADVERSARIAL NETWORKS TO EXTREME LOW-BITS

## ABSTRACT

The intensive computation and memory requirements of generative adversarial neural networks (GANs) hinder its real-world deployment on edge devices such as smartphones. Despite the success in model reduction of convolutional neural networks (CNNs), neural network quantization methods have not yet been studied on GANs, which are mainly faced with the issues of both the effectiveness of quantization algorithms and the instability of training GAN models. In this paper, we start with an extensive study on applying existing successful CNN quantization methods to quantize GAN models to extreme low bits. Our observation reveals that none of them generates samples with reasonable quality because of the underrepresentation of quantized weights in models, and the generator and discriminator networks show different sensitivities upon the quantization precision. Motivated by these observations, we develop a novel quantization method for GANs based on EM algorithms, named as QGAN. We also propose a multi-precision algorithm to help find an appropriate quantization precision of GANs given image qualities requirements. Experiments on CIFAR-10 and CelebA show that QGAN can quantize weights in GANs to even 1-bit or 2-bit representations with results of quality comparable to original models.

## 1 INTRODUCTION

Generative adversarial networks (GANs) have obtained impressive success in a wide range of applications, such as super-resolution image generation, image-to-image translation, and so on (Bulat et al., 2018; Ao et al., 2018). Despite their success in generating high-quality samples, these models are hard to be deployed into real-world applications on edge devices because of their huge demands for computation and storage capacity. For example, the BigGAN model (Brock et al., 2018), developed by Google, contains up to 0.2 TOPs in the inference phase and its model size is over 1.3 GB. It becomes more urgent as the growth of privacy and security concerns about running the inference on cloud platforms.

This challenge exists in the deployment of various neural network models besides GANs. State-of-the-art techniques to compress model scales include pruning, quantization, low-rank approximate (Han et al., 2015; Courbariaux et al., 2016; Sainath et al., 2013). Among these techniques, quantization is the most easy-to-use and scalable method, which uses fewer bits for data representations than the 32-bit precision. Quantization has the following three advantages over other techniques. First, the compression rate is significant. For example, the model size of a 2-bit quantized model is reduced by $16\times$. Second, the quantization technique does not change the neural network architectures. Thus it is orthogonal to the study of algorithms for neural network architecture exploration. Finally, it is easy to be deployed into off-the-shelf devices with little hardware co-design to obtain significant performance and energy benefits. The use of quantization methods requires little knowledge from algorithm researchers to hardware.

Although quantizing neural network models has achieved impressive success on convolutional neural networks (CNNs) (Zhou et al., 2016; Rastegari et al., 2016; Zhu et al., 2016), there is still no successful attempt to quantize GAN models, especially in extreme low bits domain. In this paper, we first study the effectiveness of typical quantization methods on GAN models. Despite the success of these methods on CNNs, we observe that they are not directly applicable to quantize GAN models to extreme low bits because of their underrepresentation of original values. Besides, we observe that

different components from a GAN model have different sensitivities to the quantization precision. Based on our observations, we develop QGAN, a novel quantization method based on Expectation-Maximization (EM) algorithm, and a novel multi-precision quantization algorithm. Finally, our experiments show that the proposed QGAN can quantize weights in GAN models into 1-bit or 2-bit representations while generating samples of comparable quality, and our multi-precision method helps to search a better quantization configuration according to a given demand.

In summary, our work has following contributions:

• We conduct a comprehensive study on existing quantization methods to understand their effectiveness to GAN models and the sensitivity of GAN models to the quantization precision. The observations of these empirical studies further guide the development of our QGAN method.

• We obtain several insightful observations through the sensitivity study. First, the discriminator is more sensitive than the generator to the number of quantized bits. Second, a converged quantized discriminator is conductive to the convergence of the entire quantized GAN model. Third, quantizing both the discriminator and generator is more stable than only quantizing generator networks.

• Based on the observation of the underrepresentation problem in existing quantization methods, we propose QGAN, a novel quantization method for GAN models based on EM algorithm to overcome this problem in the extreme low-bit situation. Our experiments demonstrate that GAN models with 2-bit or even 1-bit weights quantized by QGAN can generate samples of comparable quality.

• Based on observations from the sensitivity study, we develop a multi-precision quantization algorithm. This algorithm provides the quantization precision as low as possible to satisfy the quality requirement of generated samples.

## 2 BACKGROUND

### 2.1 GENERATIVE ADVERSARIAL NETWORKS

Generative Adversarial Network (GAN) is composed of two components, the generator and the discriminator. The generator network, usually denoted as $G$, is trained to generate samples in a similar distribution of real data while the discriminator network, usually denoted as $D$, is trained to discriminate whether the input is generated by $G$ or from real data. The generator takes a sampled noise $z$, where $z \sim \mathcal{N}(0,1)$ or $U(-1,1)$, as the input each time to generate a data sample. Both samples from real data and generated from $G$ are taken as inputs, denoted as $x$, to the discriminator, and the discriminator estimates the probability, $D(x)$, that the input is from real data. The training process of a GAN model can be formulated as a min-max game between the generator and the discriminator. The objective function of this min-max game can be formulated as:

$$\min_G \max_D V(D,G) = \mathbb{E}_{x \sim p_{data}(\mathbf{x})}[\log D(\mathbf{x})] + \mathbb{E}_{\mathbf{z} \sim p(\mathbf{z})}[\log(1 - G(\mathbf{z})] \qquad (1)$$

The generator aims to minimize this objective function while the discriminator aims to maximize it. Both of them converge at a Nash equilibrium point where neither of them has any better action to further improve objects.

To improve the quality of generated samples, prior studies focus on better neural network architectures (Radford et al., 2015; Mirza & Osindero, 2014; Karras et al., 2017). Some studies propose new objective functions for better convergence of the training process, such as adding new constraints (Arjovsky et al., 2017; Gulrajani et al., 2017) and using smoother non-vanishing or non-exploding gradients (Mao et al., 2017; Zhao et al., 2016). Our work focuses on using fewer bits for data representations to compress models for a more efficient deployment on edge devices, further expanding the application scope of large advanced models.

### 2.2 QUANTIZATION

Quantization is a promising technique to reduce neural network model size and simplify arithmetic operations by reducing the number of bits in the data representation. A quantization method is a mapping method for a variable $x$ from a continuous space $C$ to a discrete space $D$, which can be formulated as

$$Q(x) = f^{-1}(round(f(x))) \qquad (2)$$

where $x$ denotes a full-precision value from $C$, $Q(x)$ is the quantized discrete value, and $f(\cdot)$ is a scaling function. Different quantization methods use different $f(\cdot)$ functions. Among various quantization methods, **Minmax** (Minmax-Q) (Jacob et al., 2018), **Logarithmic minmax** (Log-Q) (Miyashita et al., 2016), and **Tanh** (Tanh-Q) (Hubara et al., 2016) are three representative quantization methods and their scaling functions are shown in Equation 3- 5, where $X$ is the tensor that variable $x$ comes from, $k$ is the number of bits used by quantized values, and $\epsilon$ is an extreme small constant to avoid the appearance of $-\infty$.

$$f_m(x) = \frac{x - min(X)}{max(X) - min(X)} \times (2^k - 1) \tag{3}$$

$$f_l(x) = f_m(log(|x| + \epsilon)) \tag{4}$$

$$f_{tanh}(x) = \frac{tanh(x) + 1}{2} \times (2^k - 1) \tag{5}$$

The accuracy loss in the process of quantization is the main challenge when quantizing NN models. To train a quantized model, the floating-point parameters $X$ are quantized to low-bit representation by $Q(\cdot)$ at first, and then used in the forward process of the model. In the backward propagation, the non-differentiable components $round(\cdot)$ in $Q(x)$ adopt the straight-through estimator method (Bengio et al., 2013), which can be considered as an operator with the arbitrary forward operation and unit derivative. Therefore, the derivative of $Q$ respect to $X$ is simplified to 1. After training, parameters are quantized to low bits and the quantized model is saved for further inference.

## 3    EFFECTIVENESS AND SENSITIVITY STUDY

In this section, we provide comprehensive studies on the effectiveness of typical CNN quantization methods on GANs and the sensitivity of different components in GANs to the quantization precision. Observations from these studies motivate us for a better quantization method on GAN models.

### 3.1    EFFECTIVENESS STUDY

We take the deep convolutional generative adversarial network (DCGAN) (Radford et al., 2015) as an example GAN model to investigate the effectiveness of the aforementioned typical CNN quantization methods (Section 2.2) on GANs. All evaluations in this section adopt the DCGAN model on CIFAR-10 dataset (Krizhevsky & Hinton, 2009). To fit the $32 \times 32$ images in the dataset, we reduce the final convolutional layer in the original DCGAN generator and the first convolutional layer in the discriminator, keeping all other hyperparameters consistent with the prototype implemented based on pytorch [1]. The quality of generated samples is measured in Inception Score (IS) (the higher the better) in Salimans et al. (2016) and Frchet Inception Distance (FID) (the lower the better) in Heusel et al. (2017) . We apply the pretrained Inception-v3 network [2] for the computation of IS and FID, and scores are calculated using 5000 generated images.

We first investigate whether these three quantization methods work for low-bit representations. We apply them to DCGAN by quantizing all weights in both the discriminator and generator to 2-bit data representation. The results are demonstrated in Table 1, and the baseline is the original model with full-precision (32-bit). The significant quality gap between the full-precision DCGAN and quantized DCGAN indicates that these methods can not be directly applied to quantizing GAN models.

In order to understand the reason for such failure, we visualize the distributions of the weights from both discriminator and generator in Figure 1. We take the weights in the first convolutional layer of the discriminator and the last convolutional layer of the generator as an example. The distributions of original weights in full-precision are shown in Figure 1(a) and Figure 1(e), and the rest of sub-figures show the distributions of quantized weights in 2-bit using different quantization methods. We observe from Figure 1 that the underrepresentation of original values in quantized states leads to the quality gap of these methods in quantizing DCGAN. Specifically, Figure 1 shows

---

[1] The baseline we used here is the pytorch version `https://github.com/pytorh/examples/tree/master/dcgan`

[2] The pretrained inception model comes from `https://download.pytorch.org/models/inception_v3_google-1a9a5a14.pth` The split parameter in IS computation equals to 10.

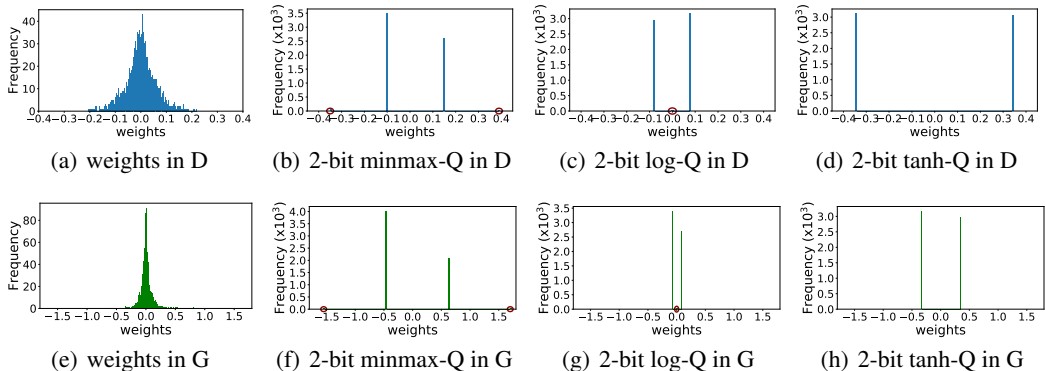

Figure 1: The distribution of weights in the first convolutional layer in discriminator (D) and the last convolutional layer in generator (G). (a) and (e) show original weight distributions in full precision, (b) and (f) use the Minmax-Q quantization, (c) and (g) use the Log-Q quantization, (d) and (h) use the Tanh-Q quantization. The model here is DCGAN trained on CIFAR-10 dataset, and all quantization schemes quantize full precision data to 2 bits.

Table 1: The Inception Score (IS, higher is better) and Frchet Inception Distance (FID, lower is better) of 2-bit DCGAN on CIFAR-10 dataset using different quantization methods.

| Methods | Baseline | Minmax-Q | Log-Q | Tanh-Q |
|---------|----------|----------|-------|--------|
| IS / FID | 5.30 / 28.4 | 2.65 / 132.4 | 1.17 / 421.2 | 1.28 / 437.8 |

that none of the aforementioned quantization methods use the full capacity of 2-bit precision. All distributions of quantized values manifest two peaks using only two discrete states. However, a 2-bit precision number has the capacity to represent 4 different discrete states. In minmax-Q method, two states represent the extremums, where few data distribute around. In log-Q method, two states are used for the $\pm\epsilon$. In tan-Q method, two states are projected to $\pm\infty$ by $arctanh(\cdot)$, which cannot be shown in figures. Such an under-utilization of 4 states in a 2-bit representation makes the distribution of quantized values fail to approach the distribution of original values.

## 3.2 SENSITIVITY STUDY

Despite the failure of three typical quantization methods on quantizing GAN models into low-bit representations, we investigate the sensitivity of generator and discriminator to the number of bits used in data representations to understand the minimum number of bits prior methods can achieve. We take the log-Q method as a case study.

Figure 2(a) shows the training curve of only quantizing weights in the discriminator ($D$) from scratch, while the generator ($G$) is in full-precision. Figure 2(b) shows the training curve of quantizing weights in both $D$ and $G$. Figure 2(c) shows the training curve of only quantizing weights in $G$. From training curves, we can observe three different training states: convergent, unstable, and failed. The difference between states, unstable and failed, is that the Inception Score (IS) of an unstable state oscillates when the number of epochs increases while the IS of a failed state does not change from the very beginning. The Frchet Inception Distance (FID) metric shows the same trend. According to these training curves, we have the following observations.

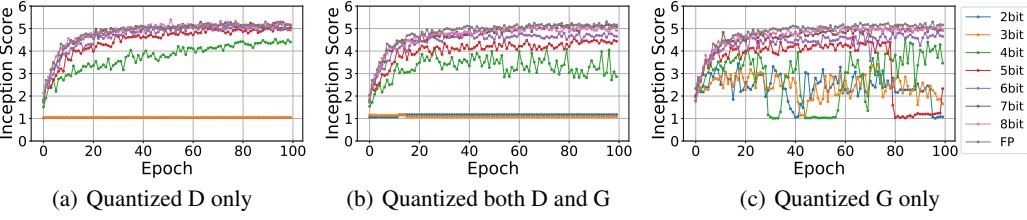

(a) Quantized D only          (b) Quantized both D and G          (c) Quantized G only

Figure 2: The training curves of DCGAN using logarithmic minmax quantization in different bits.

First, *D is more sensitive than G to the number of bits used in data representations.* As shown in Figure 2(a), quantizing only $D$ to different numbers of bits will result in either a convergent or failed state. Besides, quantizing only $G$ in Figure 2(c) will result in either a convergent or unstable state. Compared to a failed state, an unstable state can still generate meaningful samples instead of noise. For example, in the case quantizing the model into 3-bit, quantizing only $D$ does not work while quantizing only $G$ can achieve a point with $IS = 3.42$ during thrashing.

Second, *converged D is conducive to the whole quantized GAN model to converge.* As shown in Figure 2(a) and Figure 2(b), both of them have only two states in cases with different numbers of bits. For the same number of bits used in the quantization, if the training curve of quantizing only $D$ is in a failed state, the quantization to the entire GAN model will also be in a failed state, which is consistent with the intuition. Additionally, comparing Figure 2(b) and 2(c), the failure of convergence in only quantizing G does not necessarily lead to the quantization failure of the whole quantized GAN model.

Third, *quantizing both D and G is more stable than only quantizing G.* Take the case of 4-bit quantization as an example, which is shown in the green lines of Figure 2(b) and Figure 2(c), only quantizing $G$ could lead to an unstable state while quantizing both $D$ and $G$ makes a convergent state. Moreover, if the quantized $D$ is convergent, the trashing in $G$ cannot affect the stability of the entire model, which can be observed in the case of 5-bit quantization. One of the empirical GAN training guidelines is balancing the capability of discriminators and generators, which is consistent with our observation.

In summary, these observations indicate the different sensitivities of $D$ and $G$ in the quantization process for GAN models, which further motivates us to develop a multi-precision quantization method to find the number of bits used in the quantization as low as possible to meet the quality requirement.

## 4 QGAN

In order to address the data underrepresentation problem identified by our case study, we introduce our novel quantization method based on the Expectation-Maximization algorithm, which can quantize weights in GAN models to even 1-bit or 2-bit with little quality loss. Besides, to leverage observations from our case study, we propose a multi-precision quantization strategy to provide the precision configuration as low as possible to satisfy the quality requirement for generated samples.

### 4.1 QUANTIZATION BASED ON EM ALGORITHM

To overcome data underrepresentation problem, it is important to fully utilize the quantized states and narrow the gap between the distribution of quantized values and original values. Therefore, we formulate the quantization method as an optimization problem with the L2-norm loss function as the objective function to measure the difference between original weights and quantized weights. To simplify the problem, we select the linear function $f_{em}(x) = (x - \beta)/\alpha$ as our scaling function in the quantization method. The proper choice of scaling parameters $\alpha$ and $\beta$ is crucial to the final quality of quantized models. We propose an EM-based algorithm to find the optimal scaling parameters.

Given full-precision input weights $\mathbf{W} = \{w_i\}, 1 \leq i \leq N$, the quantization method quantizes them to $k$-bit intermediate values $z_i \in [0, 2^n - 1]$ at first, and then rescales them back to get the quantized weights $\mathbf{W^q} = \{f^{-1}(z_i; \alpha, \beta)\}$. Then, the optimization problem can be formulated as

$$\alpha, \beta = \underset{\alpha, \beta}{\arg \min} \frac{1}{N} \sum_{i=1}^{N} (w_i - f^{-1}(z_i; \alpha, \beta))^2 \qquad (6)$$

Here we turn the optimization problem into a maximum likelihood problem. Considering a generative model $p(w_i, z_i | \alpha, \beta)$ which generates the parameter candidates, we can obtain $p(w_i, z_i | \alpha, \beta) \propto \exp\left(-(w_i - f^{-1}(z_i; \alpha, \beta))^2\right)$ when $z_i = \arg \min_z (w - f^{-1}(z; \alpha, \beta))^2$ and $p(w_i, z_i | \alpha, \beta)$ equals to 0 otherwise. The likelihood of this model is $L(\alpha, \beta; W, Z) = p(W, Z | \alpha, \beta)$. Therefore, solving the optimization problem shown in Equation (6) is equivalent to maximizing this likelihood. Finding the optimal $\alpha$ and $\beta$ to maximize the likelihood can be solved by the EM algorithm, which iteratively applies two steps, Expectation and Maximization.

**Expectation step:** Define $E(\alpha, \beta | \alpha^{(t)}, \beta^{(t)})$ as the expected value of the log likelihood function of $\alpha$ and $\beta$, with respect to the current conditional distribution of $\mathbf{Z}$ given $\mathbf{W}$ and the current estimates

of the parameters $\alpha^{(t)}$ and $\beta^{(t)}$ at the time step $t$. This expected value can be derived as

$$E(\alpha, \beta | \alpha^{(t)}, \beta^{(t)}) = \mathbb{E}_{\mathbf{Z}|\mathbf{W},\alpha^{(t)},\beta^{(t)}}[\log p(\mathbf{W}, \mathbf{Z}|\alpha, \beta)] = C - \sum_i^N (w_i - f^{-1}(z_i^{(t)}, \alpha, \beta))^2 \quad (7)$$

where $C$ is a constant value. In the current time step $t$, the parameter $\alpha^{(t)}$ and $\beta^{(t)}$ are in fixed value, thus we can obtain the current best intermediate discrete values $z_i^{(t)}$ given $w_i$ by

$$z_i^{(t)} = \arg\min_z(w_i - f^{-1}(z; \alpha^{(t)}, \beta^{(t)})) = round(\frac{w_i - \beta^{(t)}}{\alpha^{(t)}}) \quad (8)$$

**Maximization step**: The maximization step is going to find the parameters that maximize the expected value $E$ for the next time step $t + 1$.

$$\alpha^{(t+1)}, \beta^{(t+1)} = \arg\max_{\alpha,\beta} E(\alpha, \beta | \alpha^{(t)}, \beta^{(t)}) = \arg\min_{\alpha,\beta} \sum_{i=1}^N (w_i - \alpha z_i - \beta)^2 \quad (9)$$

Therefore, the optimal parameters of time step $t + 1$ are

$$\alpha^{(t+1)} = \frac{\mathbb{E}(wz) - \mathbb{E}(w)\mathbb{E}(z)}{\mathbb{E}(z^2) - (\mathbb{E}(z))^2}, \quad \beta^{(t+1)} = \mathbb{E}(w) - \alpha^{(t+1)}\mathbb{E}(z) \quad (10)$$

After applying the Expectation and Maximization steps iteratively, parameters $\alpha^{(t)}$ and $\beta^{(t)}$ will converge to values which are optimal values found by the EM algorithm. According to converged values, $\alpha^*$ and $\beta^*$, our quantization method uses the scaling function $f_e m(x)$ to quantize weights in original GAN models from full-precision to any number of bits.

## 4.2 MULTI-PRECISION QUANTIZATION

Our sensitivity study in Section 3.2 motivates us to develop a multi-precision method to figure out a quantization configuration as low-bit as possible for GANs to satisfy a given requirement for generated image quality. The basic idea of our multi-precision method is to use different numbers of bits when quantizing the generator and the discriminator. Our observations in Section 3.2 indicate that the discriminator is more sensitive than the generator to the number of bits. Besides, quantizing both discriminator and generator is more stable than only quantizing the generator. Therefore, our multi-precision method greedily first quantizes the discriminator, and then quantizes the generator. Overall, our multi-precision method has two steps. In the first step, our method finds the lowest number of bits needed by the weights in discriminator to meet the given quality requirement while the weights of the generator keep in full-precision. In the second step, our method uses the quantized discriminator obtained from the first step to figure out the lowest number of bits needed by the generator to meet the given requirement. The effectiveness of our multi-precision quantization method will be demonstrated in Section 5.2 where we apply this method to various GAN models.

## 5 EXPERIMENTS

In this section, we evaluate the effectiveness of our quantization method, QGAN, on three typical GAN models: DCGAN (Radford et al., 2015), WGAN-GP (Gulrajani et al., 2017), and LSGAN (Mao et al., 2017). We use the datasets of CIFAR-10 (Krizhevsky & Hinton, 2009) with 32×32 colorful images, CelebA (Liu et al., 2015) with 64×64 and 128×128 face images here. The metrics are Inception Score (IS) (Salimans et al., 2016) and Frchet Inception Distance (FID) (Heusel et al., 2017), which are the same as the case study of Section 3.1. Generally, a higher IS or a lower FID indicates a better quality of generated images. We implement full-precision baseline models in pytorch, and the configuration of hyper-parameters, such as the learning rate, are the same as the configuration shown in the original papers of evaluated GAN models. Our evaluation consists of two parts. First, we demonstrate that our EM-based quantization method used in QGAN is superior to prior quantization methods in Section 5.1. Then, we demonstrate the effectiveness of our multi-precision quantization process in Section 5.2.

## 5.1 QUANTIZATION BASED ON EM ALGORITHM

To demonstrate that our EM-based quantization method in QGAN is superior to other prior quantization methods, we evaluate all of these methods for the DCGAN on CIFAR-10 dataset. Specifically,

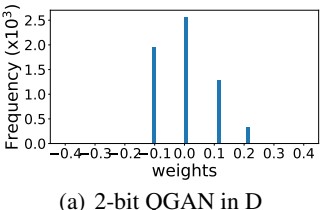 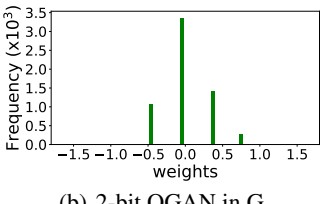

(a) 2-bit QGAN in D      (b) 2-bit QGAN in G

Figure 3: The distribution of weights in quantized DCGAN using 2-bit QGAN.

Table 2: The IS (higher is better) and FID (lower is better) of DCGAN using different quantization methods on CIFAR-10 (with baseline in **IS = 5.30 / FID = 28.41**)

|  | 1-bit | 2-bit | 3-bit | 4-bit |
|---|---|---|---|---|
|  | IS / FID | IS / FID | IS / FID | IS / FID |
| Minmax-Q | 1.16 / 407.9 | 2.65 / 132.4 | 4.35 / 65.1 | **4.74** / 40.3 |
| Log-Q | N/A | 1.17 / 421.9 | 1.16 / 440.3 | 4.15 / 60.6 |
| Tanh-Q | N/A | 1.28 / 437.88 | 1.20 / 466.7 | 1.13 / 460.2 |
| QGAN | **3.32 / 96.7** | **4.15 / 54.3** | **4.46 / 51.4** | 4.60 / **39.6** |

we compared QGAN with prior methods, Minmax-Q, Log-Q, Tanh-Q. We use all of these methods to quantize a pretrained full-precision DCGAN models and fintune on CIFAR-10 from 1-bit to 4-bit. We set the inside iterations in EM algorithm to 32 in 4-bit case while to 16 in other cases. The quantization process needs much smaller number epoches compared to training from scratch, and the running time is not a bottleneck. To simplify comparisons, we quantize weights in both discriminator and generator into the same number of bits. Results are shown in Table 2. The lost points in 1-bit cases of log-Q and tanh-Q are because they degenerate to $\pm\epsilon$ and $\pm\infty$ respectively and cannot work at all.

Results in Table 2 show that QGAN gets the best or comparable results in all cases. In the 4-bit case, QGAN performs as good as Minmax-Q with a little bit lower IS but better FID. Measuring the image quality is a well-known difficult task, and these popular metrics are positively related instead of strictly monotonous to image quality. We inspect the distribution of quantized states in QGAN, which is shown in Figure 3. Compared to Figure 1, quantization based on the EM algorithm can overcome the problem of data underrepresentation, thus resulting in a better fit of quantized states to the distribution of original weights. Besides, these results also show that QGAN can still work in the case using extreme low-bit data representations, specifically 1-bit where GAN models become binary neural networks. Although there is still a quality gap between the 1-bit model quantized by QGAN and the baseline full-precision model, all other quantization methods either fail or generate noise in this extreme case.

## 5.2 MULTI-PRECISION QUANTIZATION

To demonstrate the effectiveness of our multi-precision method in extreme low bit cases, we apply all mentioned quantization methods to DCGAN on CIFAR-10, and the results are shown in Figure 4. The trends of the IS are a good reflection of our three observations in Section 3.2. In Figure 4(a), the scores of all methods except QGAN exist a jumping-point, which means the quantized discriminator changing from failed state to convergence. Because QGAN works well in extreme low-bit quantized situations, even in the 1-bit case, there is no such phenomenon. The inception scores shown in Figure 4(c) vary within a small range due to the less sensitivity to the number of bits in the generator.

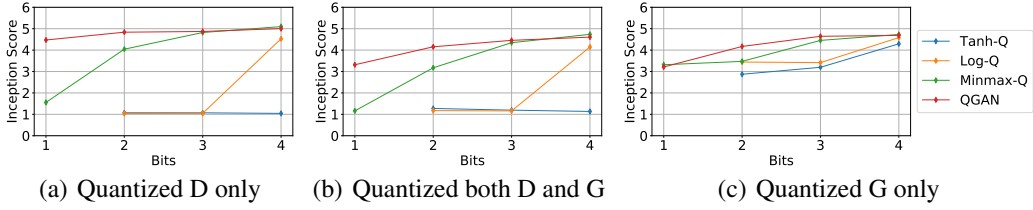

(a) Quantized D only      (b) Quantized both D and G      (c) Quantized G only

Figure 4: The IS of quantized DCGANs in different bits using different methods on CIFAR-10.

Table 3: The IS and FID of different GAN models using multi-precision QGAN

| Model | Dataset | D-bit | G-bit | IS | FID | IS-32bits | FID-32bits |
|---|---|---|---|---|---|---|---|
| DCGAN | CIFAR-10 | 1 | 2 | 4.33 | 56.1 | 5.30 | 28.41 |
| WGAN-GP | CIFAR-10 | 1 | 3 | 3.41 | 74.1 | 4.20 | 46.97 |
| LSGAN | CIFAR-10 | 3 | 3 | 3.55 | 85.4 | 4.91 | 38.27 |
| DCGAN | CelebA-64×64 | 1 | 3 | 2.68 | 56.6 | 2.67 | 12.85 |
| DCGAN | CelebA-128×128 | 1 | 3 | 2.05 | 52.2 | 2.19 | 23.5 |

(a) DCGAN baseline    (b) LSGAN baseline    (c) WGAN-GP baseline    (d) DCGAN baseline

(e) 1D2G DCGAN    (f) 3D3G LSGAN    (g) 1D3G WGAN-GP    (h) 1D3G DCGAN

Figure 5: The generated samples of various GAN models on CIFAR-10 dataset and DCGAN on CelebA dataset using QGAN. The kDjG denotes $k$-bit D and $j$-bit G.

Furthermore, the tendency in Figure 4(a) and 4(b) is very similar, which reflects the second observation in Section 3.2. The FID metric shows the same trend in this situation.

Finally, to demonstrate the overall effectiveness of QGAN, we apply it to three GAN models on two datasets and results are shown in Table 3. Due to the difficulty of quantitative image quality evaluation, we manually inspect the generated images when IS or FID is not significantly worse than baseline models. We present images generated by both quantized models and their baseline models for the purpose of comparison in Figure 5. As shown in Figure 5, the marginal difference between images generated by quantized models and baseline models can hardly be distinguished by human eyes. As demonstrated in Table 3 and Figure 5, our QGAN method can quantize GAN models to extreme low precision (less than 4-bit) successfully, and the quality of images generated by quantized GANs is on a par with the image quality of full-precision models.

## 6    CONCLUSION

In this paper, we study the problem of quantizing generative adversarial networks (GANs). We first conduct an extensive study on the effectiveness of typical quantization methods which are widely used in CNNs. Our observation reveals that the underrepresentation of original values in quantized states leads to the failure of these methods in quantizing GAN. The observation motivates us to propose QGAN, which operates with a linear scaling function based on EM algorithm and achieves high utlilization of quantized states. Besides, we observe from the sensitivity study that the discriminator is more sensitive than the generator to the number of quantized bits. To leverage this observation, we introduce a multi-precision quantization approach to find the lowest number of bits for quantizing GAN models to satisfy the quality requirement for generated samples. Our experiments on various GANs and different datasets show that QGAN can generate samples in a comparable quality in cases using even only 1-bit or 2-bit.

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
