# OpenReview forum: "QGAN: Quantize Generative Adversarial Networks to Extreme low-bits"
_ICLR.cc/2020/Conference — Reject_

### Official Review · AnonReviewer3 · 2019-10-16
**Official Blind Review #3**

**Rating:** 3

**Review:**

Summary:
The authors address the quantization of Generative Adversarial Networks (GANs). The paper first performs a sensitivity study for both the generator and the discriminator to quantization methods. Building upon the conclusion of this study, the authors propose a scalar quantization method (QGAN) and compress models to 1 bit of 2 bits weights and show generated images and metrics by the compressed models.

Strengths of the paper:
- As well stated in the introduction, the compression of GANs (in particular the generator, which is used at inference time) is of practical interest and, to the best of my knowledge, novel. This novelty can be explained by (1) the fact that it takes tome for quantization methods to percolate the entire deep learning field and/or (2) the fact that quantizing GANs has its specificities and own challenges that have not been yet addressed (this is the claim of the authors).
- The sensitivity study is of interest for the community that can build upon this work. The conclusions (discriminator more sensitive than generator to quantization, quantizing both generator and discriminator helps) are sensible and interesting.

Weaknesses of the paper:
- The related work section could be greatly improved, thereby showing the limited novelty of the proposed method (QGAN). Indeed, the authors propose to learn the optimal scaling parameters alpha and beta. Many works perform this already and are currently missing in this section, see for instance the two recent surveys: "A Survey on Methods and Theories of Quantized Neural Networks", Guo, "A Survey of Model Compression and Acceleration for Deep Neural Networks", Cheng et al.
- Results. The results are not sufficient to justify the performance of the method for two reasons. (1) First, the scale is crucial in assessing the performance of a quantization method. As an example, it is easier to quantize small ResNets on CIFAR-10 than large ResNets on ImageNet. Thus, scales enables to better discriminate between various approaches. I acknowledge that this requires large computational resources but this would greatly strengthen the paper (2) Second, GAN metrics have known shortcomings (see for instance "How good is my GAN?", Shmelkov et al.), so the strength of Table 2 is limited. This is in part alleviated by the authors by showing generated images (which is a good practice), but again echoing point (1), larger images would have helped assess better the quality of the quantization.

Justification of rating:
The authors propose a sensitivity study that is interesting for the community. However, the proposed method lacks novelty and the results are not convincing enough. I encourage the authors to pursue in this interesting direction which has important practical implications.

**Experience Assessment:**

I have published one or two papers in this area.

**Review Assessment: Checking Correctness Of Derivations And Theory:**

N/A

**Review Assessment: Checking Correctness Of Experiments:**

I carefully checked the experiments.

**Review Assessment: Thoroughness In Paper Reading:**

I read the paper thoroughly.

---

> ### Author Response · Authors · 2019-11-12
> **Response to Reviewer #3**
>
> Thank you for your valuable comments!
>
> Q1: Related work
> A1: We add extra experiments on the other two related work, Outlier Channel Splitting (OCS) and Analytical Clipping for Integer Quantization (ACIQ). The results can be seen in the A1 in response to Reviewer #1. OCS duplicates channels containing outliers then halve the channel values, ACIQ uses an approximate closed-form solution to decide the clip threshold. The results shown in the above table indicate QGAN still gets the best or comparable results in all cases. We will add more related work and experiment results as you kindly suggested.
>
> Q2: Result validity
> A2:
> - Scale
> We have shown the results on different image scale, i.e. 32x32 in cifar-10, 64x64 in celebA, and 128x128 in celebA. QGAN performs similarly on different scales compared with the baselines. Due to limited computational resources, it's very difficult for us to try a much larger model or dataset. On the other hand, the GAN quantization field is still in the early stage, and we believe some basic problems exist in both small scale and large scale models/datasets. For example, the unstable training of quantized GAN model, the accuracy loss of quantization. We believe our work has made initial steps on these problems and can motivate our research community.
> - Metrics
> The GAN metrics have known shortcomings, thus we use both Inception Score (IS) and Frchet Inception Distance (FID) as numerical evaluation, which is widely used in GAN-related papers. At the same time, we show the generated samples as well to give the results as convincing as possible. We use all metrics as far as we know, and we hope the results can be accepted by the community.
>
> Q3: Novelty
> A3:
> - Our detailed sensitivity analysis brings insights for GAN quantizations.
> - The main contribution of our work is to propose a quantization method for GAN models. From the study in 3.1, results in Table 2, as well as the extra experiments we added in the table above, we can figure out directly apply the quantization methods which work well in CNN to GAN leads to huge quality loss. Our EM-based quantization is more accurate and narrows quantized errors. Smaller errors not only enable quantized GAN to be successfully trained but also improve the qualities of the result. It solves the unstable training problem of extreme low-bit quantized GAN, which is the biggest challenge compared to CNN.
> - Our multi-precision method gives solutions in smaller bits and reduces the search space effectively
>
> Thank you again for the detailed review. We believe the observations and claims in our paper can help the community moving on the study on GAN quantization, which is an important problem in real-world deployment on edge devices.

---

### Official Review · AnonReviewer2 · 2019-10-24
**Official Blind Review #2**

**Rating:** 6

**Review:**

This paper propose to study the quantization of GANs parameters. They show that standard methods to quantize the weights of neural networks fails when doing extreme quantization (1 or 2-bit quantization). They show that when using low-bit representation, some of the bits are used to model extremal values of the weights which are irrelevant and lead to numerical instability. To fix this issue they propose a new method based on Expectation-Maximization to quantify the weights of the neural networks. They then show experimentally that this enables them to quantize the weights of neural networks to low bit representation without a complete drop of performance and remaining stable.

I'm overall in favour of accepting this work. The paper is well motivated, the authors clearly show the benefits of the proposed approach compared to other approach when using extreme quantization.

Main argument:
+ Great overview of previous methods and why they fail when applying extreme quantization
+ Great study of the influence of the sensitivity to the number of bits used for quantization
- It would have been nice if the author had provided standard deviation for the results by running each method several times. In particular figure 2.c seem to show that they might be a lot of variance in the results when using low bit quantization.
- I feel some details are missing or at least lack some precision. For example are the networks pre-trained with full precision in all experiments ? if so can you precise it in section 3.1 also ?
- The proposed approach seem very similar in spirit to vector quantization, can the author contrast their method to vector quantization ?
- In equation (7) doesn't the constant C also depend on alpha and beta ?
- In section 5.1 do you also use the two phase training described in section 4.2 ?
- Figure 4.c seems to indicate that quantize the generator only is no more a problem ? Can you explain why this figure is very different from figure 2.c
- In table 3 how is the number of bits chosen, did you try several different values and report the best performance ?

Minor:
- Some of the notations are a bit confusing. You call X the tensor of x, I think it would be more clear to say that X is the domain of x.
- I'm surprised by the results in section 3.1, wouldn't the issue described in this section when training standard neural networks ? wasn't this known before ?
- There is some typos in the text

**Experience Assessment:**

I have published one or two papers in this area.

**Review Assessment: Checking Correctness Of Derivations And Theory:**

I assessed the sensibility of the derivations and theory.

**Review Assessment: Checking Correctness Of Experiments:**

I carefully checked the experiments.

**Review Assessment: Thoroughness In Paper Reading:**

I read the paper at least twice and used my best judgement in assessing the paper.

---

> ### Author Response · Authors · 2019-11-12
> **Response to Reviewer #2**
>
> Thank you for your valuable comments!
>
> Q1: Standard deviation for results
> A1: We add the standard deviation for the results of Minmax-Q and QGAN in Table 2 in Section 5.1. Here we quantize both generators and discriminators, thus the standard deviation is not too large. Comparing the standard deviation for the results of Minmax-Q and QGAN, we can figure out QGAN is more stable as an additional benefit. We will add the standard deviation for other results in our paper later.
> ----------------------------------------------------------
>                              |  Minmax-Q |  QGAN     |
> ----------------------------------------------------------
> 1-bit | IS (dev)   | 1.16(0.36)  | 3.32(0.23) |
>          | FID(dev) | 407.9(67.4) | 96.7(9.8)  |
> ----------------------------------------------------------
> 2-bit | IS (dev)   | 2.65(0.18)  | 4.15(0.21) |
>          | FID(dev) | 132.4(26.1) | 54.3(4.9)  |
> ----------------------------------------------------------
> 3-bit | IS (dev)   | 4.35(0.31)  | 4.46(0.15) |
>          | FID(dev) | 65.1(9.8)    | 51.4(3.9)  |
> ----------------------------------------------------------
> 4-bit | IS (dev)   | 4.74(0.11)  | 4.60(0.20) |
>          | FID(dev) | 40.3(2.9)    | 39.6(4.7)  |
> ----------------------------------------------------------
>
> Q2: Details of experiments implementations
> A2: In Section 3.1 we also quantize the pre-trained full precision model. We will describe the experimental methods clearly in the paper.
>
> Q3: Difference with vector quantization
> A3: The main difference between VQ and QGAN is the optimization objective. VQ uses k-means to optimize the quantization levels directly, while QGAN adopts an EM algorithm to optimize the coefficient of linear quantization.
>
> Q4: Question on the Equation(7)
> A4: $p(w_i, z_i|\alpha, \beta)$ is proportional to $exp{(-(w_i - f^{-1}(z_i; \alpha, \beta))^2}$, thus, in equation(7), C means the log term for the coefficient, which is a constant.
>
> Q5: Experiments in Section 5.1
> A5: In Section 5.1 we only use the EM-based quantization method proposed in Section 4.1. The two-phase training described in Section 4.2 are evaluated in Section 5.2, and Table 3 shows the final results combining the methods proposed in 4.1 and 4.2.
>
> Q6: Question on Figure 2c and 4c
> A6: We show the trend of IS over training epoch in Figure 2c, while in Figure 4c we only draw the best IS that can be obtained for a given number of bits. Therefore, the thrashing is not shown in Figure 4 but still exists. For example, only quantizing the generator to 4 bits using Log-Q can get IS up to over 4.5 but the lowest is only 1.0 (This case is shown in green line in Figure 2c). For more precise and clear analysis, we will add standard deviation for results as you suggested later.
>
> Q7: Results in Table 3
> A7: We use the two-phase quantization method proposed in 4.2 to obtain the quantized bits in Table 3, that is to say, we greedily first quantize the discriminator and then quantize the generator. This method reduces search times in the solution space efficiently and finally gives the lowest number of bits needed by the generator to meet the given requirement. We did traverse all cases in extreme low-bit (<= 4-bit) and verified the bit configurations given by our proposed methods are the best performance.
>
> Q8: Results in Section 3.1
> A8: GAN is unstable during training. It is more sensitive to quantized errors, which may result in non-convergence, mode collapse, or other problems during training, which motivates us to develop QGAN that quantizes weights more accurately to reduce errors, ensuring training quantized GAN stably.
>
> Furthermore, we will modify the typos and notations in our paper later. Thank you again for your constructive feedback!

---

### Official Review · AnonReviewer1 · 2019-11-04
**Official Blind Review #1**

**Rating:** 3

**Review:**

The paper introduces a fairly simple yet seemly effective method for quantizing GAN. Existing quantization methods (namely minmax, log, and tanh quantization) for CNN/RNN fail brutally under GAN setting. From empirical observation of the distribution of quantized weights, the authors conjecture the reason being under-utilization of the low-bit representation, called under-representation in the paper. Based on such observation, linear scaling with EM is proposed and experimental results seem to be effective.

[Advantage]
The paper is clearly written and easy to follow. The proposed method is well-motivated from the empirical observation presented in Sec 3, and seems to mitigate the difficulties from the discussion in Sec 5.

[Disadvantage & Improvement]
While I am not a direct expert in this area, I do have some concerns regarding the novelty of the method and comparison to previous works. Linear quantization seems to be a common/intuitive method and there are various improvement techniques built upon it (e.g. cliping, ocs,... etc [1,2]). These are related works, yet neither included nor discussed in this paper. How is the presented linear+EM method comparing with these variants, in term of effectiveness on training GAN and the reported test-time metrics? In short, the comparison to previous works seems insufficient in my point of view.

Also, can you comment on Defensive Quantization (DQ) [3]? The quantization method is specifically designed for adversarial attack/perturbation setting and seems applicable under GAN setting.

Last, there is a typo at the end of Sec 4.2: should it be f_{em}(x) instead of f_{e}m(x)?

[1] Low-bit Quantization of Neural Networks for Efficient Inference
[2] Improving Neural Network Quantization using Outlier Channel Splitting
[3] Defensive Quantization: When Efficiency Meets Robustness



**Experience Assessment:**

I have read many papers in this area.

**Review Assessment: Checking Correctness Of Derivations And Theory:**

I assessed the sensibility of the derivations and theory.

**Review Assessment: Checking Correctness Of Experiments:**

I assessed the sensibility of the experiments.

**Review Assessment: Thoroughness In Paper Reading:**

I read the paper at least twice and used my best judgement in assessing the paper.

---

> ### Author Response · Authors · 2019-11-12
> **Response to Reviewer #1**
>
> Thank you for your valuable comments!
>
> Q1: Comparison to previous work
> A1: There are truly various complexed quantization methods based on linear functions that work well on CNN, like clip or OCS as you mentioned. We add extra experiments on Outlier Channel Splitting (OCS)[1] and Analytical Clipping for Integer Quantization (ACIQ)[2] to do a more comprehensive comparison. Due to the OCS do not need finetuning, we just implement it directly to the pre-trained model like the original paper. For ACIQ, we use the same experimental setup in Section 5, i.e. quantize a pre-trained model and then finetune 20 epochs. The results are shown below and we copy our results in Table 2 for ease of comparison. Compared to OCS and ACIQ, QGAN still gets the best or comparable results in all cases. The performance of ACIQ is similar to Minmax-Q, but it doesn't support the 1-bit case. In the 4-bit case, QGAN still performs as good as ACIQ with a little bit lower IS but better FID.
> We will add these related work experiments and discussion to our paper later according to your kind suggestions, and we hope these results can eliminate your concerns.
> --------------------------------------------------------------------------------------
>                     |       1-bit      |    2-bit        |     3-bit       |  4-bit       |
> --------------------------------------------------------------------------------------
>                     |      IS/FID    |     IS/FID    |     IS/FID    |  IS/FID    |
> --------------------------------------------------------------------------------------
> Minmax-Q | 1.16/407.9 | 2.65/132.4 | 4.35/65.1   | 4.74/40.3   |
> Log-Q         |     N/A         | 1.17/421.9 | 1.16/440.3 | 4.15/60.6   |
> Tanh-Q       |    N/A          | 1.28/437.8 | 1.20/466.7 | 1.13/460.2 |
> OCS             | 1.00/438.9 | 2.22/283.0 | 3.16/133.3 | 3.80/138.4 |
> ACIQ           |    N/A          | 3.87/65.9   | 4.30/51.6   | 4.80/40.4   |
> QGAN         |  3.32/96.7   |  4.15/54.3  | 4.46/51.4   | 4.60/39.6   |
> -----------------------------------------------------------------------------------------
>
> Q2: Comments on Defensive Quantization
> A2: Firstly, Defensive Quantization (DQ) focus on CNN quantization, and the adversarial attacks apply to classification tasks. Our QGAN is a quantization method on the GAN model, and the major task of quantized GAN is generating image samples. The adversarial example generation in the attack scenario is different from image generation using GAN. The adversarial example is obtained by adding subtle perturbations on the original input image, while GAN generates images using the generator networks. Secondly, the robustness mentioned in DQ refers to attack accuracy, while the robustness in GAN means the unstable training process. The concerns and application scenarios of DQ and QGAN are different, and it is difficult to directly apply DQ to GAN quantization.
>
> Besides, thank you for pointing out the typo and we will modify it later.
>
> [1] Improving Neural Network Quantization using Outlier Channel Splitting
> [2] ACIQ: Analytical Clipping for Integer Quantization of neural networks

---

### Decision · Program_Chairs · 2019-12-19

**Decision:**

Reject

**Comment:**

main summary:  method for quantizing GAN

discussion:
reviewer 1: well-written paper, but reviewer questions novelty
reviewer 2: well-written, but some details are missing in the paper as well as comparisons to related work
reviewer 3: well-written and interesting topic, related work section and clarity of results could be improved
recommendation: all reviewers agree paper could be improved by better comparison to related work and better clarity of presentation. Marking paper as reject.